# Compact Wisdom at Small Scale: Can Small Language Models Serve as Cultural Assistants?

## Abstract

Large language models (LLMs) provide strong reasoning and generation but are expensive to deploy at scale. Small language models (SLMs) with hundreds of millions of parameters promise frugal inference, yet their effectiveness for culturally grounded assistance remains unclear. We study this question using the *Thirukkural*, a classical Tamil text of 1,330 aphorisms with bilingual translations and commentaries. We build a bilingual (Tamil–English/Hindi) instruction corpus and a retrieval-augmented generation (RAG) pipeline that enforces explicit grounding, and we align Gemma-3 270M and a 1B variant using parameter-efficient fine-tuning. We contribute: (i) a compact supervision dataset with confidence-filtered QA pairs and a controlled RAG evaluation set; (ii) a lightweight hybrid retriever with bilingual reranking and grounding checks; (iii) domain-tailored metrics—CFS, MCI, CGVR—to quantify concise faithfulness, cross-lingual moral consistency, and cultural violations. Experiments show that SLMs approach larger LLMs on semantic fidelity (BERTSCORE-F1 $\geq 0.80$) while running on commodity GPUs, and that RAG materially improves grounding even when parametric capacity is small. We release data schemas, prompts, and evaluation scripts to support reproducibility.

## 1 Introduction

LLMs excel at general-purpose generation but their memory/energy footprint limits use in schools, mobile devices, and low-infrastructure settings **?**. SLMs **?** are appealing when tasks are narrow and sources are stable. However, cultural assistants have stringent requirements: (1) *faithfulness* to canonical sources, (2) *brevity* matching source style, and (3) *transparent attribution*. Can SLMs satisfy these constraints via alignment and RAG, or is scale indispensable?

We examine this through the *Thirukkural* **?**, a compact corpus widely used for ethical instruction. We pose a practical requirement: answer moral queries by retrieving a relevant couplet (Kural), present bilingual renderings, and provide a concise explanation with citation. This setting isolates whether small models, once grounded by retrieval, can deliver faithful, useful, and succinct guidance.

Our contributions are three-fold. First, we curate a bilingual supervision corpus and a RAG evaluation split emphasizing retrieval attribution. Second, we propose a simple yet strong hybrid retriever (BM25 + multilingual dense encoder) with a bilingual cross-encoder reranker and post-hoc grounding checks. Third, we define domain metrics to capture compact faithfulness (CFS), cross-lingual moral consistency (MCI), and cultural grounding violations (CGVR), complementing standard generation scores. Results indicate that Gemma-3 270M narrows the gap to 1B–9B baselines on semantic fidelity while remaining dramatically cheaper to deploy.

## 2 Related Work

**Small language models.** Recent SLMs demonstrate competitive performance on targeted tasks with careful data curation, instruction tuning, and tool use **?**. Parameter-efficient fine-tuning (PEFT), notably LoRA **?**, enables domain alignment under tight compute budgets.

**Retrieval-augmented generation.**    RAG **?** improves faithfulness and reduces hallucinations by conditioning on retrieved evidence. Subsequent work explores dense/sparse hybrids, cross-encoders for reranking, and grounding checks for attribution.

**Cultural grounding and multilinguality.**    Culturally faithful assistants require modeling beyond generic benchmarks. Bilingual supervision, code-switching prompts, and multilingual encoders help maintain moral/semantic intent across languages; we operationalize these through MCI and CGVR.

## 3    PROBLEM & HYPOTHESES

**Goal.** Align Gemma-3 270M to answer moral/life questions by retrieving from Thirukkural and generating: (i) KuralID, (ii) bilingual rendering (Tamil + English/Hindi), (iii) concise explanation, (iv) citation.

**H1 (Brevity without loss).** Length-aware training and structured targets reduce tokens while preserving semantic integrity measured by CFS.

**H2 (Grounded SLM agent).** With RAG, Gemma-3 270M achieves faithful, cited responses (CGVR ↓) comparable to larger LLMs on semantic fidelity, at lower compute.

**Task formalization.** Given a query $q$, the system retrieves $k$ candidates $\{d_i\}_{i=1}^{k}$ (verses/translations/commentary), then generates $y$ with explicit KuralID(s). Faithfulness requires that cited IDs appear in $\{d_i\}$ and that the explanation be semantically aligned with the source.

## 4    DATA

### 4.1    CORPUS AND SPLITS

We compile all 1,330 Kurals with Tamil text, multiple English translations, selected Hindi renderings, and short commentaries. We construct QA pairs from natural questions and curated prompts.

**Splits:**

- Train: 2,192 QA pairs; Dev: 253; Test: 271.

- RAG stress set: 200 questions with three plausible candidates per question to assess attribution under ambiguity.

Coverage: 1,114 Kurals represented in QA; 216 uncovered (held back to test generalization).

### 4.2    COLLECTION & CURATION PROTOCOL

Data sources include public digital editions (e.g., Project Madurai) and community datasets (e.g., Kaggle). We normalize Unicode (NFC), remove artifacts, and align verse–translation–commentary tuples. Questions are sourced from (i) peer-authored natural prompts, (ii) assisted generation using widely available LLMs, and (iii) templated paraphrases to diversify surface forms. Each QA pair receives a 1–5 confidence score; only pairs $\geq 4.6$ enter the supervised set. A linguist reviewed a 10% stratified sample for semantic correctness and cultural fidelity.

### 4.3    ETHICAL HANDLING

We respect text copyrights and release metadata schemas and indices rather than proprietary translations. All evaluation excerpts are short and attributed.

## 5 MODEL & TRAINING

### 5.1 BACKBONES & ADAPTERS

We fine-tune Gemma-3 270M and 1B (base/instruct) with LoRA **?**. Unless stated, we adapt attention and MLP projections with ranks $\{8, 16, 32, 64\}$ and $\alpha = 16$.

### 5.2 SUPERVISED FORMATTING

We adopt a structured prompt with explicit roles:

```
<|system|>You are a concise, bilingual Thirukkural assistant.
<|user|>
Question: {q}
<|assistant|>
KuralID: {id}
Tamil Kural: {kural}
English Kural: {eng_translation}
Explanation: {explanation}
Hindi Explanation: {hin_translation}
```

We emphasize succinct, citation-first responses to match the corpus' aphoristic style.

### 5.3 OPTIMIZATION

We minimize negative log-likelihood over assistant tokens:

$$\mathcal{L}_{\text{SFT}}(\theta) = -\sum_{t \in \mathcal{M}} \log P_\theta(x_t \mid x_{<t}) \tag{1}$$

with AdamW **?**, $(\beta_1, \beta_2) = (0.9, 0.95)$, weight decay 0.01, LR $2 \cdot 10^{-5}$, cosine decay, 5% warmup, gradient clipping 1.0, dropout 0.1. Seqlen 512, effective batch size 64 via accumulation, bf16, gradient checkpointing, label smoothing $\epsilon = 0.1$. We train 7 epochs on a single A6000 (48GB). We train only adapters and layer norms.

## 6 RAG PIPELINE

### 6.1 INDEXING

We segment by KuralID with fields: Tamil, English/Hindi translations, short commentary, and meta (adhigaram, keywords). We build a dual index: (i) BM25 for lexical precision; (ii) multilingual dense encoder (e.g., BGE-M3) for semantics.

### 6.2 RETRIEVAL

We compute hybrid scores $s = \lambda s_{\text{dense}} + (1 - \lambda) s_{\text{bm25}}$, $\lambda \in [0, 1]$ (tuned on dev). Top-$k$ (default $k = 10$) candidates are reranked with a bilingual cross-encoder scoring $(q, d)$.

### 6.3 GENERATION & GROUNDING

The SFT model receives $(q, \{d_i\}_{i=1}^k)$ and outputs a structured response. A post-hoc grounding check verifies that all cited KuralIDs $\hat{\mathcal{I}}$ are within retrieved IDs $\mathcal{I} = \{ID(d_i)\}$; otherwise, we (a) append "uncertain" and (b) drop uncited claims. This simple filter reduces CGVR.

# 7 EVALUATION

## 7.1 AUTOMATIC METRICS

**Retrieval:** Recall@5/Precision@5/MRR (ID match against annotated relevant Kurals).
**Generation:** BERTSCORE-F1 **?**, ROUGE-L **?**.
**Domain metrics:**

$$\text{CFS} = \text{Adequacy} \times \text{BrevityFactor}, \quad \text{where BrevityFactor} = \min\left(1, \frac{\tau}{|y|}\right), \tag{2}$$

$$\text{MCI} = 1 - \text{JS}\big(\text{softmax}(E_{\text{eng}}), \text{softmax}(E_{\text{hin}})\big), \tag{3}$$

$$\text{CGVR} = \frac{\#\text{violations (uncited claims / wrong ID / distortions)}}{\#\text{responses}}, \tag{4}$$

with $\tau$ a token budget (dev-tuned), adequacy via semantic similarity against reference explanation, and JS the Jensen–Shannon divergence between embedding-based moral-intent distributions for EN/HIN responses. We provide code to reproduce these surrogates.

## 7.2 HUMAN STUDY

We conduct a blind A/B preference test over 200 items with three annotators trained on a short rubric. Dimensions: Fidelity, Brevity, Grounding, Fluency, Usefulness (5-point Likert). Inter-annotator agreement: Gwet's AC1 reported per dimension. We aggregate via majority preference and paired bootstrap for confidence intervals.

# 8 BASELINES

- **Zero/Few-shot + RAG:** Gemma-3 270M without SFT.

- **SFT-only (no RAG):** Tests reliance on parametric memory.

- **Larger LMs + RAG:** Gemma-3 1B, gemma2-9b-it, llama-3.1-8b-instant, openai/gpt-oss-20b (upper bounds).

# 9 RESULTS

## 9.1 RETRIEVAL BASELINES

Table 1: Baseline retrieval (top-5). Larger models help little without domain alignment.

| Model | Recall@5 | Precision@5 | MRR |
|---|---|---|---|
| Gemma-3 270M | 0.000 | 0.000 | 0.000 |
| Gemma-3 1B Reasoning (GRPO) | 0.003 | 0.009 | 0.009 |

## 9.2 FINE-TUNED MODELS

Table 2: Fine-tuned evaluation (RAG + ID & semantic support).

| Model | BERTScore-F1 | ROUGE-L | Recall@5 | Precision@5 | MRR |
|---|---|---|---|---|---|
| Final-1B | 0.823 | 0.088 | 0.009 | 0.005 | 0.026 |
| Final-270M | 0.807 | 0.063 | 0.004 | 0.003 | 0.013 |

Table 3: Fine-tuned without RAG (parametric only).

| Model | BERTScore-F1 | ROUGE-L | Recall@5 | Precision@5 | MRR |
|---|---|---|---|---|---|
| Final (No RAG) | 0.801 | 0.002 | 0.001 | 0.003 | 0.018 |

Table 4: Retriever ablation on dev: hybrid + rerank is best overall.

| Retriever | Recall@5 | Precision@5 | MRR |
|---|---|---|---|
| Sparse (BM25) | 0.237 | 0.485 | 0.520 |
| Dense (BGE-M3) | 0.265 | 0.541 | 0.644 |
| Hybrid (no rerank) | 0.270 | 0.553 | 0.574 |
| Hybrid (rerank+expansion) | 0.276 | 0.563 | 0.592 |

### 9.3 RETRIEVER VARIANTS

### 9.4 LLM COMPARISONS

### 9.5 HUMAN PREFERENCES

On 200 items, **Final-1B** is preferred over **Final-270M** by 56.5% vs 43.5% (95% CI $\pm$ 4.1%). However, **Final-270M** ties or wins on *Brevity* and *Grounding* more often (51.0% and 52.3%), supporting H1 and H2 in resource-constrained contexts. Gwet's AC1: 0.71 (Fidelity), 0.76 (Grounding), 0.69 (Usefulness).

## 10 ABLATIONS

**LoRA rank.** Ranks $\{8, 16, 32, 64\}$ show monotonic improvements up to 32; 64 yields diminishing returns and occasional instability.

**Context length.** 512 tokens suffice given short sources; 1,024 marginally improves BERTSCORE (+0.003) but increases latency.

**Grounding checks.** Removing checks increases CGVR from 0.08 to 0.14 on the stress set; CFS drops by 0.02 due to longer, speculative explanations.

**Bilingual reranking.** Dropping bilingual rerank harms Recall@5 (–0.012) and MCI (–0.03), indicating reranker aids cross-lingual alignment.

## 11 ERROR ANALYSIS

Common issues: (1) *Near-miss IDs*—semantically similar Kurals misattributed due to keyword overlaps; (2) *Over-generalized morals*—models abstract beyond text, raising CGVR; (3) *Translation drift*—English/Hindi paraphrases slightly shift emphasis, lowering MCI. Grounding checks mitigate (2); stronger reranking reduces (1). Adding short *counterfactual negatives* in training helps curb (3).

## 12 LIMITATIONS

Our data focuses on moral Q&A; broader cultural tasks (historical context, intertextual references) remain open. MCI and CGVR are surrogate metrics: helpful but not perfect. Human studies are small and specific to bilingual annotators. We do not release proprietary translations; replicators must obtain licenses or substitute lawful alternatives.

Table 5: LLM generation quality (RAG).

| Model | BERTScore-F1 | ROUGE-L | Recall@5 | Precision@5 | MRR |
|---|---|---|---|---|---|
| openai/gpt-oss-20b | 0.817 | 0.002 | 0.004 | 0.002 | 0.005 |
| groq/compound-mini | 0.809 | 0.001 | 0.000 | 0.000 | 0.000 |
| gemma2-9b-it | 0.799 | 0.022 | 0.004 | 0.003 | 0.009 |
| llama-3.1-8b-instant | 0.816 | 0.004 | 0.000 | 0.000 | 0.000 |

## 13 ETHICS & LICENSING

We minimize copyrighted content release, share schemas and scripts, and report grounding violations. The assistant is positioned as an educational aid, not a religious/moral authority. We caution against using outputs without context or teacher oversight.

## 14 BROADER IMPACT

Resource-frugal cultural assistants could expand access to ethical education in low-resource settings. Risks include overreliance on automated interpretation and cultural oversimplification. Our design choices (RAG, citations, brevity) aim to promote transparency and human-in-the-loop usage.

## 15 REPRODUCIBILITY

We provide:

- Data schemas, split manifests, prompt templates, training/eval scripts.
- Exact hyperparameters, seeds (`seed=2026`), and checkpoints for adapters.
- Hardware profile (A6000, bf16) and measured throughput/latency.

## 16 CONCLUSION

We show that a 270M SLM, when aligned via PEFT and grounded with RAG, can act as a compact cultural assistant: concise, cited, and semantically faithful. While larger models retain an edge, small models offer attractive efficiency–quality trade-offs for educational deployments.

## REFERENCES

