# OpenReview forum: "Compact Wisdom at Small Scale: Can Small Language Models Serve as Cultural Assistants?"
_ICLR.cc/2026/Conference — ICLR 2026 Conference Withdrawn Submission_

### Official Review · Reviewer_PcAa · 2025-10-27

**Soundness:** 1
**Presentation:** 1
**Contribution:** 1
**Rating:** 0
**Confidence:** 5

**Summary:**

This paper curate a bilingual supervision corpus and a RAG evaluation benchmark for Tamil.

This paper is incomplete with many empty sections and zero references. It is impossible to evaluate this paper. The authors should withdraw it.

**Strengths:**

None

**Weaknesses:**

This paper is incomplete with many empty sections and zero references. The authors should withdraw it.

**Questions:**

The authors should withdraw this paper.

---

### Official Review · Reviewer_ojmM · 2025-10-29

**Soundness:** 2
**Presentation:** 2
**Contribution:** 2
**Rating:** 2
**Confidence:** 2

**Summary:**

This paper investigates whether small language models (SLMs) can function as culturally grounded assistants through retrieval-augmented generation (RAG) and lightweight alignment. Using the Thirukkural, the authors build a bilingual (Tamil–English/Hindi) QA corpus, a hybrid retrieval system, and domain-specific metrics (CFS, MCI, CGVR) to measure faithfulness, moral consistency, and cultural grounding. Experiments show that a 270M-parameter Gemma-3 model fine-tuned with LoRA and RAG achieves semantic fidelity comparable to much larger models while running efficiently on a single GPU, demonstrating that compact, well-grounded SLMs can deliver concise, faithful, and transparent cultural guidance.

**Strengths:**

- The work curates a carefully filtered Tamil–English/Hindi QA dataset with explicit retrieval attribution and bilingual evaluation for cross-lingual moral reasoning and cultural grounding.
- The paper defines new metrics (CFS, MCI, and CGVR) to quantify brevity, moral alignment, and citation accuracy.

**Weaknesses:**

- The evaluation is restricted to Thirukkural-based moral reasoning and bilingual QA, which despite of cultural richness represents a narrow domain. It’s unclear whether the proposed framework generalizes to other cultural corpora, genres, or non-aphoristic texts.
- The introduced metrics (CFS, MCI, CGVR) prioritize conciseness and citation accuracy, potentially penalizing models that produce culturally rich but paraphrased or interpretive responses. This biases the evaluation toward literal faithfulness rather than nuanced understanding.
- The paper does not directly compare small-model performance against compact LLMs (e.g., Mistral-7B, Phi-3-mini) or retrieval-tuned baselines of similar scale. This makes it difficult to quantify how much of the gain stems from the proposed grounding methods versus dataset design.

**Questions:**

- Did the authors test variants without reranking, without post-hoc grounding, or without LoRA fine-tuning to measure the contribution of each?
- Could the authors provide more details of how human raters or experts filtered or rated the data (e.g., consistency criteria, inter-annotator agreement)?
- Could the authors provide validation results, e.g., correlation of the proposed metrics with human judgments, to justify that they reflect faithfulness and moral alignment rather than stylistic similarity.
- How does the retrieval or reranker balance multiple valid references, and handle conflicting or multi-faceted moral interpretations when multiple verses could support different moral outcomes?
- Did the authors test MCI with different multilingual encoders to check robustness? The method for measuring moral alignment across languages (MCI) could be influenced by embedding choice.

**Details Of Ethics Concerns:**

Although the paper emphasizes educational uses, deploying such a model without adequate disclaimers might lead users, especially students, to interpret its responses as normative moral advice. This raises fairness and societal responsibility concerns.

---

### Official Review · Reviewer_oYB8 · 2025-10-29

**Soundness:** 1
**Presentation:** 1
**Contribution:** 2
**Rating:** 2
**Confidence:** 1

**Summary:**

This paper investigates an interesting and socially relevant topic: the feasibility of using Small Language Models (SLMs) with Retrieval-Augmented Generation (RAG) to act as compact, faithful, and cross-lingual cultural assistants, exemplified by the classical Tamil text, the Thirukkural. The work contributes a novel supervision dataset, a hybrid retriever, and domain-specific metrics (CFS, MCI, CGVR).

While the research motivation is commendable and the framework for developing domain-tailored metrics is innovative, the experimental design and interpretation of results suffer from critical flaws that severely undermine the paper's core claims regarding the effectiveness of SLMs in this role. Specifically, the low faithfulness scores and the failure to establish a fair comparison with larger models negate the central hypothesis.

**Strengths:**

1. The exploration of resource-frugal cultural assistants addresses a key limitation of LLMs (cost/deployment footprint) and highlights a socially valuable application in low-infrastructure settings.

2. The introduction of CFS (Compact Faithfulness), MCI (Cross-Lingual Moral Consistency), and CGVR (Cultural Grounding Violation Rate) is a significant methodological contribution. These metrics go beyond standard generation scores to quantify the specific requirements of cultural assistance (brevity, faithfulness, and moral alignment).

**Weaknesses:**

1. The experimental setup fails to establish a fair performance ceiling, making the claim that SLMs "approach larger LLMs"  highly questionable

2. The paper claims that SLMs can be faithful and concise. However, the automatic evaluation metrics strongly contradict this claim:

Extremely Low ROUGE-L: The ROUGE-L scores for both the Final-270M (0.063) and Final-1B (0.088) models (Table 2) are extremely low, even when paired with RAG11. ROUGE-L measures literal overlap and structural fidelity against the reference text12. A score this low implies that the generated "concise explanation" (which should match the source style 13) deviates drastically in wording and structure from the gold reference, directly contradicting the requirement for brevity matching source style and faithfulness

Contradictory Metrics: The high BERTScore-F1 ($\approx 0.80$) 15 (semantic similarity) combined with low ROUGE-L (literal similarity) suggests the models grasp the general meaning but are incapable of producing the precise, aphoristic, and structure-loyal text required by the task. This undermines the argument that the SLMs provide "compact wisdom"

3. The paper does not have a reference at all.

4. The presentation and the completeness of the paper is extremely weak.

**Questions:**

See weakness.

---

### Official Review · Reviewer_iXLr · 2025-10-31

**Soundness:** 1
**Presentation:** 1
**Contribution:** 1
**Rating:** 0
**Confidence:** 5

**Summary:**

The paper studies whether small language models with a few hundred million parameters can act as “cultural assistants” when grounded with retrieval from Thirukkural, a classical Tamil text. The authors build a bilingual (Tamil–English/Hindi) instruction corpus, construct a retrieval-augmented generation pipeline, and fine-tune Gemma-3 270M and 1B models with LoRA. The system answers moral questions by retrieving a relevant couplet, presenting bilingual renderings, and producing a short explanation with citation. The work also introduces domain-specific metrics to quantify concise faithfulness, cross-lingual moral consistency, and cultural grounding violations.

**Strengths:**

The paper contributes a curated dataset centered on culturally grounded reasoning, which is underrepresented in current evaluation suites. The problem formulation is clear, and the overall pipeline—hybrid retrieval, bilingual reranking, grounding checks—addresses practical requirements (brevity, attribution, cross-lingual alignment). The authors report both automatic metrics and a small human study. The release of data schemas, prompts, and scripts improves reproducibility and may support future work on culturally aligned generation.

**Weaknesses:**

The evaluation is limited to a single source and a narrow bilingual setting. It is therefore hard to judge whether the proposed pipeline generalizes beyond Thirukkural-style moral queries.

The paper evaluates several retriever settings (BM25, dense BGE-M3, hybrid with/without reranking), but the analysis remains brief, and a broader set of dense/cross-encoder alternatives is not explored in depth. For generation, the paper reports BERTScore-F1 and ROUGE-L, but the discussion does not explain discrepancies between these metrics (e.g., cases where similar BERTScore contrasts with much lower ROUGE-L in fine-tuning with vs. without RAG). Although a human A/B study is reported, it does not target cases where BERTScore-F1 and ROUGE-L diverge; a small error/qualitative analysis for those discrepancies would aid interpretation.

Comparisons with larger LMs use off-the-shelf models under a RAG prompt (no fine-tuning). The paper does not analyze how model size interacts with retrieval quality, nor whether the approach transfers to architectures outside the Gemma family. Ablations are present (retriever variants, LoRA rank, context length, grounding checks), but they do not isolate whether alternative dense/cross-encoder choices would change outcomes; a targeted comparison would strengthen the claim.

**Questions:**

The sections on retriever variants and LLM comparisons mostly present tables of numbers without interpretation. Could the authors comment on common failure modes, and explain what could prevent further improvement?

The paper states that several fine-tuning parameters of the overall pipeline were explored. Could the authors provide more detailed validation results showing how these choices influenced retrieval quality, grounding violations, and brevity? A short summary of how the final configuration was selected would clarify the robustness of the chosen setup.

---

### Note · Authors · 2025-11-29

**Comment:**

We have not submitted the completed version. So we are withdrawing it

**Withdrawal Confirmation:**

I have read and agree with the venue's withdrawal policy on behalf of myself and my co-authors.